# A Comparative Study of the Effects of Anticoagulants on Pure Platelet-Rich Plasma Quality and Potency

**DOI:** 10.3390/biomedicines8030042

**Published:** 2020-02-25

**Authors:** Hachidai Aizawa, Hideo Kawabata, Atsushi Sato, Hideo Masuki, Taisuke Watanabe, Tetsuhiro Tsujino, Kazushige Isobe, Masayuki Nakamura, Koh Nakata, Tomoyuki Kawase

**Affiliations:** 1Tokyo Plastic Dental Society, Kita-ku, Tokyo 114-0002, Japan; sarusaru@mx6.mesh.ne.jp (H.A.); hidei@eos.ocn.ne.jp (H.K.); atu.net@nifty.com (A.S.); hideomasuki@gmail.com (H.M.); watatai@mui.biglobe.ne.jp (T.W.); tetsudds@gmail.com (T.T.); kaz-iso@tc4.so-net.ne.jp (K.I.); maoh4618@me.com (M.N.); 2Bioscience Medical Research Center, Niigata University Medical and Dental Hospital, Niigata 951-8520, Japan; radical@med.niigata-u.ac.jp; 3Division of Oral Bioengineering, Institute of Medicine and Dentistry, Niigata University, Niigata 951-8514, Japan

**Keywords:** anticoagulants, EDTA, ACD-A, heparin, platelets, platelet-rich plasma, PDGF

## Abstract

It is generally accepted that citrate or the A-form of acid-citrate-dextrose (ACD-A) are suitable for preparing platelet-rich plasma (PRP) for regenerative therapy. However, this is based on evidence from blood transfusions and not from regenerative medicine. Thus, we examined the effects of anticoagulants, such as ACD-A, ethylenediaminetetraacetic acid (EDTA), and heparin, on the regenerative quality of PRP to address this gap. The blood samples were collected in the presence of anticoagulants and were processed to prepare pure-PRP. Platelet size, activation status, and intra-platelet free Ca^2+^ concentration were determined while using a hematology analyzer and flow cytometer. Platelet-derived growth factor-BB (PDGF-BB) was quantified while using an ELISA. In pure-PRP samples, EDTA caused platelet swelling and activation, but yielded the highest number of platelets. Heparin aggregated platelets and disturbed the overall counting of blood cells. However, no significant differences in PDGF-BB levels were observed among the anticoagulants tested. Moreover, when considering the easy preparation of platelet suspensions, without the need for high-level pipetting skills, these findings suggest the comparable potency of EDTA-derived pure-PRP in tissue regeneration and support the use of EDTA in the preparation of pure-PRP. Further in vivo studies are required in animal models to exclude the possible negative effects of including EDTA in pure-PRP preparations.

## 1. Introduction

The development and clinical use of platelet-rich plasma (PRP) is undoubtedly an outstanding innovation and it has markedly impacted regenerative medicine. PRP has been increasingly used in various fields of regenerative therapy to date since the first report appeared in 1998 [1]. In parallel, several in-vitro studies that were performed by our group and others have provided evidence to support its clinical application [2,3,4,5,6,7]. However, its academic value has not been highly considered. A major reason for this paradoxical evaluation is due to the low predictable clinical outcomes of PRP. Several randomized controlled trials (RCT) have been performed while using PRP; however, all of the evaluations contradict each other, especially in bone regeneration. In addition, different or unexpected clinical outcomes have been superficially explained by individual differences in terms of both PRP potency and host site conditions. We think that there are three probable reasons for this: (1) application cases are scarcely selected, (2) PRP preparation quality is rarely assured, and (3) PRP action mode is considerably complicated to be revealed clearly.

As PRP is considered to be a type of adjuvant therapy [8], its clinical outcomes could be majorly influenced by the regenerative potential of each individual and prior or simultaneous medication or surgical treatments. These factors are rarely modified to make them compatible with PRP therapy, whereas PRP quality can be considerably optimized by clinicians in clinical settings. In this context, the term “PRP quality” that we have used in our previous and present studies can be tentatively defined as “the basic concept underlying the maximization of PRP therapy predictability”. However, it is poorly understood which factors, such as platelet counts, platelet functional activity, and inclusion of leukocytes, primarily influence “PRP quality”. Thus, further investigation should be performed to affirm or modify this definition and eventually establish a practical basis for PRP quality assurance and control. We have provided a supplemental explanation of “PRP quality” in the Discussion section.

It is difficult to perform large-scale RCTs and obtain convincing evidence without any stable quality assurance and control. Based on this concept, in the past several years, we have studied PRP preparation quality control and standardization [8,9,10,11,12]. Recently, several impact articles have been published to emphasize the necessity of PRP standardization, including platelet-rich fibrin (PRF) standardization [13,14,15,16,17,18,19,20,21,22]. However, some concerns, such as the operator skill being the most influential factor for PRP quality during manual PRP preparation, are yet to be addressed.

Several medical societies have attempted to formulate PRP preparation guidelines. Harrison et al. (2018) recently surveyed the consciousness of clinicians toward PRP preparation and therapy [23]. Clinician consciousness toward anticoagulants, especially EDTA use, exceeded expectations. Although citrate-based anticoagulants have been generally used for blood transfusion [24], and Marx (2001) and Anitua et al. (2016) recommended using citrate or A-form of acid-citrate-dextrose (ACD-A) as an anticoagulant for preparing PRP [25,26], Harrison et al. (2018) reported that ~20% subjects think that EDTA is more preferable than citrate or ACD-A for PRP preparation [23].

It is thought that anticoagulants that are suitable for PRP preparation should be minimally harmful against cells or tissues surrounding the implantation sites. In addition, such adverse effects should be reversible, even if occurring after implantation. All of the anticoagulants used in this study are used as medications in clinical settings. Thus, it can be assumed that when administered at an appropriate dose, these anticoagulants are safe and they cause no recognizable adverse effects on the surrounding tissues. In this study, we aimed to compare the quality of pure-PRP samples prepared from whole-blood samples that were anticoagulated with various anticoagulants and examined their size, activation status, and intra-platelet free Ca^2+^ concentrations using an automated hematology analyzer and a flow-cytometer (FCM). We also quantified the platelet-derived growth factor-BB (PDGF-BB) as a representative growth factor in PRP treatment while using ELISA. This study focuses on a fundamental, but probably long misunderstood, topic in regenerative dentistry.

## 2. Materials and Methods

### 2.1. Preparation of Pure-PRP

The blood samples were collected from 15 non-smoking healthy volunteers aged 26 to 72 years. Peripheral blood (~7 mL) was collected in three different plastic vacuum blood collection tubes (Venoject II; Terumo, Tokyo, Japan) containing (1) 1 mL ACD-A (2.20 *w*/*v*% sodium citrate, 0.90 *w*/*v*% citrate, 2.20 *w*/*v*% glucose) (Terumo, Tokyo, Japan), (2) EDTA powder (contained in Venoject II; Terumo), and (3) heparin-coated films (contained in Venoject II; Terumo). The whole-blood samples were gently mixed by repeated gentle agitation and then stored at 22–25 °C for up to 24 h. The whole-blood samples were then centrifuged at 472× *g* for 10 min (1st soft spin) using a centrifuge equipped with swing rotor (KS-5000; Kubota, Tokyo, Japan) at 22–25 °C and the upper plasma fractions were again centrifuged at 984× *g* for 5 min (2nd hard spin) while using a centrifuge that was equipped with an angle rotor (Sigma 1-14; Sigma Laborzentrifugen GmbH, Osterode am Harz, Germany) at 22–25 °C The platelet pellets were gently resuspended in 250 μL supernatant acellular plasma and 50 μL platelet suspension, i.e., pure-PRP, were used to determine blood cell counts. The remaining was stored at −80 °C until use for ELISA.

For FCM analysis (see below), the platelet pellets were suspended in phosphate buffer saline (PBS) containing 0.1% bovine serum albumin (BSA) to obtain a 40–60 × 10^4^ platelets/μL solution. Notably, a single operator prepared all of the PRP samples.

The ethics committee for human participants at the Niigata University School of Medicine approved the study design and consent forms for all procedures (Niigata, Japan, project code: 2297, approved on 14 October 2015) and complied with the Helsinki Declaration of 1964, as revised in 2013. We have got the written informed consent from the patient.

### 2.2. Determination of Cell Counts and Platelet Size

Platelet, white (WBC), and red blood cell (RBC) counts in whole-blood samples and PRP preparations were determined while using a pocH 100iV automated hematology analyzer (Sysmex, Kobe, Japan). In the case of ACD-A-anticoagulated samples, which were diluted by ACD-A, individual cell counts were compensated by the dilution factor (1.2). Platelet size parameters, such as platelet distribution width (PDW), mean platelet volume (MPV), and platelet-large cell ratio (P-LCR), were simultaneously determined.

### 2.3. Flow-cytometric Analysis of Surface Antigens (CD41, CD62P)

In a previous study [27], we found that 24-h storage of ACD-A-anticoagulated whole blood samples did not significantly reduce the quality of platelet-rich fibrin. Thus, we stored the whole-blood samples for up to 24 h with appropriate agitation to compare the effects of EDTA with those of ACD-A. As described previously [10,28], platelet suspensions were prepared and treated with CaCl_2_ (0.1% at the final concentration) in polypropylene sample tubes for 10 min After incubation, the platelets were fixed with ThromboFix (Beckman-Coulter, Brea, CA, USA) for 30 min The platelets were then washed with PBS and simultaneously probed with phycoerythrin-conjugated mouse monoclonal anti-CD41 and fluorescein isothiocyanate-conjugated anti-CD62P antibodies (5 μL per 100 μL sample) (BioLegend, San Diego, CA, USA) for 50 min at room temperature. Mouse IgG1 (BioLegend) was employed for isotype controls. After washing twice with PBS, the platelets were analyzed on an FCM (Cell Lab Quanta SC; Beckman-Coulter Inc., Brea, CA, USA). Data were analyzed while using the FlowJo software (FlowJo LLC, Ashland, OR, USA).

### 2.4. Flow-cytometric Analysis of Intra-Platelet Free Ca^2+^ Concentrations

As described above, 100 μL platelet suspensions were prepared from whole-blood samples that were stored with gentle agitation for 24 h at room temperature and in 0.1% BSA-containing PBS. Platelet suspensions were then treated with 5 μg/mL Fluo4-AM (Dojin, Kumamoto, Japan) [29] for 30 min and directly subjected to flow-cytometric analysis [30,31] at 488 nm/535 nm (excitation/emission).

### 2.5. Determination of PDGF-BB Levels by ELISA

Pure PRP preparations were prepared from whole-blood samples stored for 24 h and then stored at −80 °C until use. After thawing and diluting the samples with PBS, PDGF-BB concentrations were determined while using human PDGF-BB Quantikine ELISA kits (R&D Systems Inc., Minneapolis, MN, USA) [7]. We did not activate platelets before freezing in this study; however, we preliminarily conformed that the freeze-thawing process could release growth factor stored in platelets. With respect to the dilution of the samples, the PDGF-BB levels in whole-blood samples were diluted by 1.2-fold only in the case of ACD-A. Furthermore, the pure-PRP samples that were prepared from the anticoagulated blood samples were diluted by a factor of 50 with PBS.

### 2.6. Statistical Analysis

The data are expressed as the mean ± standard deviation (SD) (Figure 1 and Figure 2) or by box plots (Figure 3, Figure 4, Figure 5, Figure 6 and Figure 7). For multigroup comparisons, statistical analyses were conducted to compare the mean values by one-way analysis of variance (ANOVA), followed by Tukey’s multiple-comparison test (SigmaPlot 12.5; Systat Software, Inc., San Jose, CA, USA). Differences with *p* values of <0.05 were considered to be statistically significant.

## 3. Results

Figure 1 shows the time-course effects of anticoagulants on platelet, WBC, and RBC counts in whole-blood samples. Heparin induces platelet aggregation and adhesion to WBC; hence, platelet counts cannot necessarily be accurately determined by the automated hematology analyzer. However, we presented these data as reference for comparison. As predicted, heparin- anticoagulation treatments significantly reduced platelet counts when compared to other anticoagulant treatments at the same time points from 1 to 24 h. However, the WBC counts significantly reduced only at 24 h after heparin-anticoagulation treatments when compared to ACD-A-anticoagulation treatments. In contrast, no significant differences were observed in RBC counts among the groups.

Figure 2 shows the time-course effects of anticoagulants on platelet size (PDW, MPV, and P-LCR) in whole-blood samples. EDTA- and heparin-anticoagulation treatments significantly increased the platelet distribution width when compared to the ACD-A-anticoagulation treatments at the same time points from 1 to 24 h, among all of the parameters studied.

Figure 3 shows the platelet, WBC, and RBC counts in pure PRP preparations. EDTA- anticoagulation treatments yielded the highest counts of platelets. WBCs were not contaminated in all of the pure-PRP samples, and a significant difference in the WBC counts was observed between ACD-A and heparin-anticoagulated samples. Additionally, RBCs were not contaminated; however, EDTA–anticoagulation treatments increased RBC counts the most.

Figure 4 shows the effects of anticoagulants on platelet size in pure-PRP. The representative platelet histograms show that heparin-anticoagulation treatments increased the number of large platelets. A comparison of PDW, MPV, and P-LCR among different anticoagulated samples revealed that they were increased in EDTA- and heparin-anticoagulated samples when compared to anticoagulant-free and ACD-A-anticoagulated whole-blood samples.

Figure 5 shows the effects of anticoagulants and CaCl_2_ stimulation on CD62P expression on platelets. The representative histograms of CD62P^+^ platelets (Figure 5a) reveal that both EDTA- and heparin-anticoagulation treatments increased the CD62P^+^ platelet count in non-stimulated control platelets. CaCl_2_ stimulation also increased CD62P^+^ platelet counts (vs. control) in all of the groups. However, EDTA and heparin seemed to reduce CaCl_2_ effectiveness (Figure 5b,c). CaCl_2_ stimulation significantly increased CD41^+^, CD62P^+^ platelet counts in ACD-A- and EDTA-anticoagulated samples, but not in heparin-anticoagulated samples. However, the increase was much higher in the ACD-A-anticoagulated samples than in the EDTA-anticoagulated samples. Thus, EDTA and heparin both increased the basal (i.e., control) levels of CD62P^+^ platelets, but reduced the CaCl_2_-induced increase in CD62P^+^ platelet levels.

Figure 6 shows the effects of anticoagulants on intra-platelet free Ca^2+^ concentrations, which is thought to influence platelet activity and functions [32]. The representative histograms of Fluo4^+^ platelets reveal that the overall fluorescence intensity in EDTA-anticoagulated samples was somewhat lower than that in the ACD-A-anticoagulated samples. Moreover, the fluorescence intensity in the heparin-anticoagulated samples was the highest. The quantification of median values confirmed this order (heparin >> ACD-A > EDTA) (Figure 6b). The Fluo4^+^ platelet percentage was lower in EDTA- and heparin-anticoagulated samples than that in ACD-A-anticoagulated samples; however, the difference was not significant.

Figure 7 presents the effects of anticoagulants on PDGF-BB concentrations in pure-PRP samples, the values of which were corrected by the dilution factor. No significant difference in the PDGF-BB concentration was observed in various anticoagulated samples (Figure 7a); however, the platelet counts were the highest in EDTA-anticoagulated samples (Figure 3a). Based on these data, PDGF-BB concentration in pure-PRPs were calculated per unit platelet count (10^7^ platelets). PDGF-BB concentrations were the highest in these samples due to apparent low platelet count in heparin-anticoagulated samples, as shown in Figure 7b.

## 4. Discussion

This study examined the effects of three anticoagulants, ACD-A, EDTA, and heparin, on pure-PRP quality, i.e., platelet and growth factor contents. To date, it has been generally accepted that citrate-based anticoagulants, such as citrate and ACD-A, are suitable for preparation of PRP in regenerative therapy. However, the biomedical basis for this choice is mainly based on accumulated evidence in transfusion [24].

To the best of our knowledge, the reason for EDTA exclusion in PRP preparations has not clearly been explained previously. As described or demonstrated in previous studies [23,33,34,35], we have confirmed that EDTA significantly swelled and activated the platelets. Moreover, EDTA is more efficient than ACD-A (or citrate) for inhibiting platelet aggregation and subsequent platelet collection in PRP. The preparation of well-suspended and homogenous PRP is much easier with EDTA than with other anticoagulants, which should be considered to be an advantage of EDTA use over other anticoagulants. Thus, it is not clear whether these effects are sufficient for excluding EDTA use. We speculate that perhaps the conventional protocol for PRP preparation using citrate-based anticoagulants has been followed in the field of regenerative therapy for two major reasons: one is that this protocol has been widely and continuously used as a standard method for laboratory investigation for testing platelet aggregation and surgical procedures utilizing “fibrin glue”. The additional reason might be overconcern of EDTA’s cytotoxicity.

In our hypothesis, EDTA has been repeatedly claimed to be inappropriate for PRP preparation, as EDTA might functionally disrupt platelets by chelating Ca^2+^ involved in many functions [32], and their growth factors may be lost or reduced from PRP prepared from EDTA-anticoagulated whole-blood samples. As predicted, platelet functions were suppressed to some extent; however, their growth factors seemed to be more preserved than predicted. Accumulated data from in-vitro studies indicate that the efficacy of PRP that is used in regenerative medicine is mostly dependent on the release and quality of growth factors [5,6,33,36]. Most clinicians and researchers widely accept this concept. However, it can also be considered that the quantity and quality of growth factors are strictly related to those of platelets contained in PRP preparations. Based on this concept, we have developed simple methods for point-of-care testing and claimed their importance to assure PRP quality and improve the predictability of PRP therapy [8,12,37]. In any case, assuming that PRP clinical potency is eventually dependent on the PRP function as a carrier of growth factors, these findings suggest that EDTA might be used in PRP preparation.

As for the negative effects of EDTA, it has often been demonstrated that EDTA suppresses the regenerative activity of cells that are involved in tissue regeneration, e.g., stromal cells [22,33]. However, these findings were obtained in vitro, where the culture system is closed and the exchange of nutrients and waste is limited. Thus, EDTA might influence stromal or some other cell types in this closed environment. In contrast, EDTA must diffuse away or bind to calcium-containing components, such as hydroxyapatite in skeletal tissue, and blood is continuously circulated. Thus, it can be reasoned that EDTA does not remain at the implantation sites long enough in its free form to hamper the activity of surrounding cells. Only a few publications have suggested possible clinical applicability of EDTA-anticoagulated PRP preparations based on the results of preclinical animal or clinical studies. Tajima et al. demonstrated that EDTA-anticoagulated PRP preparations, in combination with adipose-derived stem cells, stimulate bone regeneration without apparent adverse effects [38].

Usually, PRP is prepared on-site, as needed for regenerative therapy. Thus, as a standard protocol of preparation, PRP should be derived from autologous blood and directly used without freezing or other processing that disrupts blood cells. However, it is not necessary to maintain platelets as living cells in PRP if PRP is used as a source of growth factors that are involved in tissue regeneration. If PRP, especially leukocyte-rich PRP, is also expected to kill bacteria and debride their debris, it might be required that leukocytes be maintained in PRP, as they are in our body. In this study, we confirmed that, even though their activity was suppressed to some extent, platelets were alive and maintained their functions in EDTA-prepared PRP. Since large-sized leukocyte types, which have a relatively short life in vitro [39], were countable in EDTA-anticoagulated stored whole-blood samples, we speculate that leukocytes remain alive and functional in PRP. However, the effects on leukocytes should be investigated in detail in later studies.

The limitations of this study should be noted. To date, a question as to which anticoagulants are suitable for PRP preparation in regenerative therapy has often been addressed [35,40,41]. These studies performed no animal studies to obtain evidence strongly supporting the choice of EDTA, although it is commonly concluded that EDTA is superior to other anticoagulants in yielding nonaggregate platelets. In this study, we did not, either, but we demonstrated that platelets are not severely damaged functionally after 24-h incubation. Nevertheless, to date, EDTA has not been recommended for use in PRP preparation for regenerative therapy [23]. The trend seems to be changing in PRP preparation protocols from “anticoagulant-dependent” to “no additives” for higher safety. However, it is not too late to reevaluate the choice of EDTA for the preparation of high-quality PRP.

Finally, we would like to provide a supplementary explanation regarding “PRP quality”. The term “quality” was introduced from the manufacturing perspective on the basis of the concept that cells used for regenerative therapy are biomedical products, i.e., biomedicines. In the case of most advanced cell medicinal products, such as mesenchymal stem cells, quality is generally defined by safety and efficacy, which are further classified into sterility, purity, identity, potency, and stability [9,42]. In particular, purity is considered to be the most important parameter because of the possibility of pluripotent nucleated cells giving rise to tumorigenesis. However, because of their cell contents, the possibility of this occurring with PRP is unlikely and, therefore, less attention has been paid to quality assurance. Instead, we have persistently stressed the necessity for the quality inspection of individual PRP samples for other possible parameters, such as growth factor retention and quality and quantity of platelets, prior to clinical use, to ensure and standardize PRP quality [8,9,12,37]. At minimum, all of the product inspections should be conducted immediately after application to “normalize” the clinical outcomes and thus make randomized clinical trial results more convincing. From a similar point of view, Milants et al. concurrently proposed novel classifications of PRP on the basis of blood cell content and activation status [43]. However, the current situation is not yet conducive for the establishment of a standard PRP therapy with predictable results. Standardization at the level of setting a preparation protocol, the necessity of which has been increasingly proposed, cannot fundamentally solve the problem of less-predictable clinical outcomes due to their quality variations.

## 5. Conclusions

The anticoagulant that is used for PRP preparation needs to be carefully selected to avoid platelet aggregation or damage. In this study, we demonstrated that ACD-A preserved platelet morphology (size) and functionality (activation and growth factor retention) better than other anticoagulants that are used for PRP preparation. As ACD-A has a lower chelating ability compared with EDTA [44] and has no detectable adverse effects on platelets, leukocytes, or other cells involved in tissue regeneration, it is a good choice for use as an anticoagulant. Heparin is inappropriate for PRP preparation and therapy due to the difficulties that are inherent in quality control and retrieval of coagulation activity. Historically, EDTA has also been considered to be inappropriate for PRP preparation. However, we found that EDTA can simplify the preparation of well-suspended PRP and preserve PDGF-BB levels to a similar extent as ACD-A. Therefore, if EDTA doses served by PRP preparations are maintained at sub-toxic topical levels at implantation sites, these in-vitro data suggest that EDTA should be reconsidered as a promising alternative to citrate-based anticoagulants for PRP preparation.

## Figures and Tables

**Figure 1 biomedicines-08-00042-f001:**
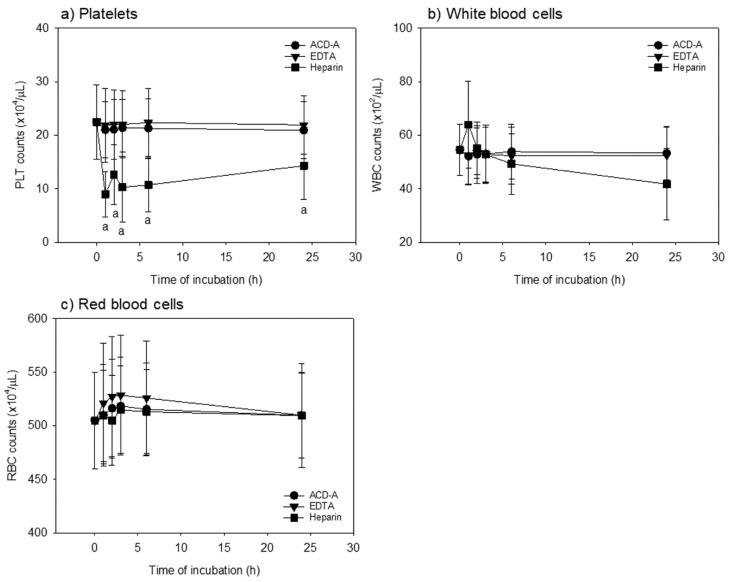
Time-course effects of anticoagulants on platelet and white and red blood cell counts in whole-blood samples. Whole-blood samples were stored following mild mixing at room temperature for up to 24 h. (**a**) Platelet counts: heparin-anticoagulation treatments significantly (*p* < 0.05, vs. A-form of acid-citrate-dextrose (ACD-A) and EDTA) reduced platelet counts (vs. other anticoagulation treatments) at the same time points from 1 to 24 h. (**b**) White blood cell counts: heparin-anticoagulation treatments significantly reduced white blood cell counts only at 24 h (vs. ACD-A-anticoagulation treatments). (**c**) Red blood cell counts: no significant differences were observed among the groups. *n* = 15.

**Figure 2 biomedicines-08-00042-f002:**
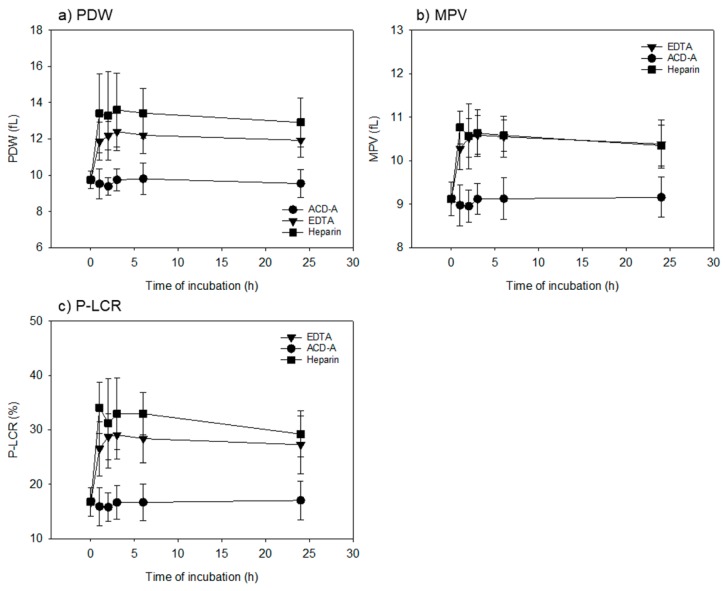
Time-course effects of anticoagulants on platelet size (platelet distribution width, mean platelet volume and platelet-large cell ratio) in whole-blood samples. Whole-blood samples were stored following mild mixing at room temperature for up to 24 h. (**a**) platelet distribution width. (**b**) Mean platelet volume. (**c**) platelet–large cell ratio. In all the parameters, EDTA and heparin anticoagulation treatments significantly increased platelet distribution width (vs. ACD-A anticoagulation treatments) at the same time points from 1 to 24 h. *n* = 15.

**Figure 3 biomedicines-08-00042-f003:**
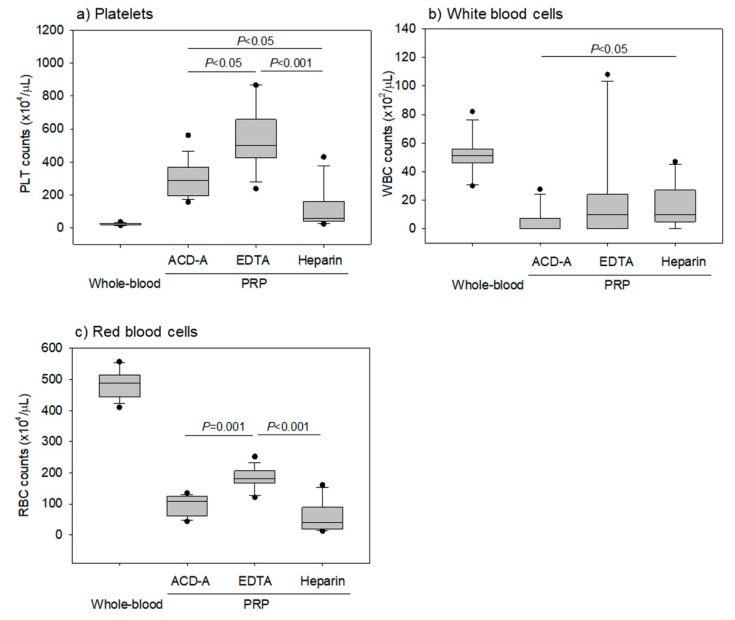
Counts of platelets, white blood cells, and red blood cells in pure-platelet-rich plasma (PRP) preparations. Pure-PRP samples were prepared from whole-blood samples stored for 24 h. (**a**) Platelet counts. (**b**) White blood counts. (**c**) Red blood cell counts. *n* = 15.

**Figure 4 biomedicines-08-00042-f004:**
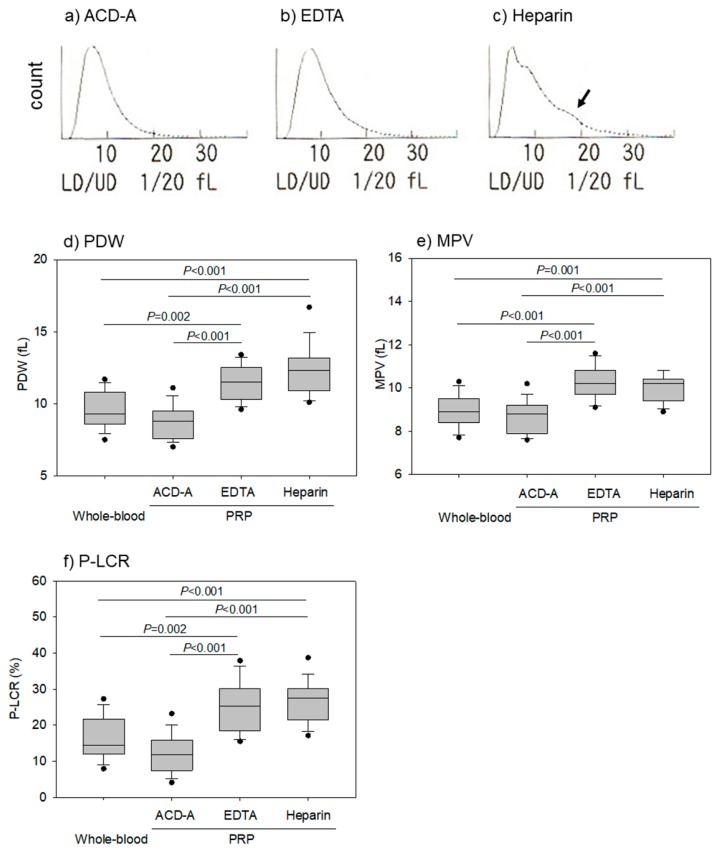
Effects of anticoagulants on platelet size in pure-PRP. Pure-PRP samples were prepared from whole-blood samples stored for 24 h. (**a**–**c**) Representative platelet histograms. (**d**–**f**) Parameters of platelet size. *n* = 15. An arrow in panel c represents platelet detected as “large platelet”.

**Figure 5 biomedicines-08-00042-f005:**
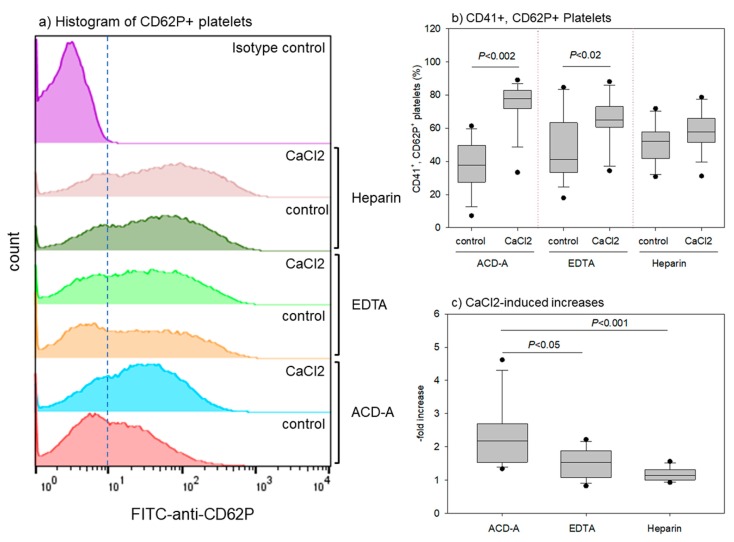
Effects of anticoagulants and CaCl_2_ stimulation on CD62P expression in platelets. Pure-PRP samples were prepared from whole-blood samples stored for 24 h. Platelet suspensions in 0.1% BSA-supplemented PBS were stimulated with 0.1% CaCl_2_ and fixed platelets were subjected to flow-cytometrical analysis. (**a**) Representative histograms of CD62P^+^ platelets. (**b**) Percentages of CD41^+^, CD62P^+^ platelets. (**c**) CaCl_2_-induced increases in the double positive platelets. *n* = 14.

**Figure 6 biomedicines-08-00042-f006:**
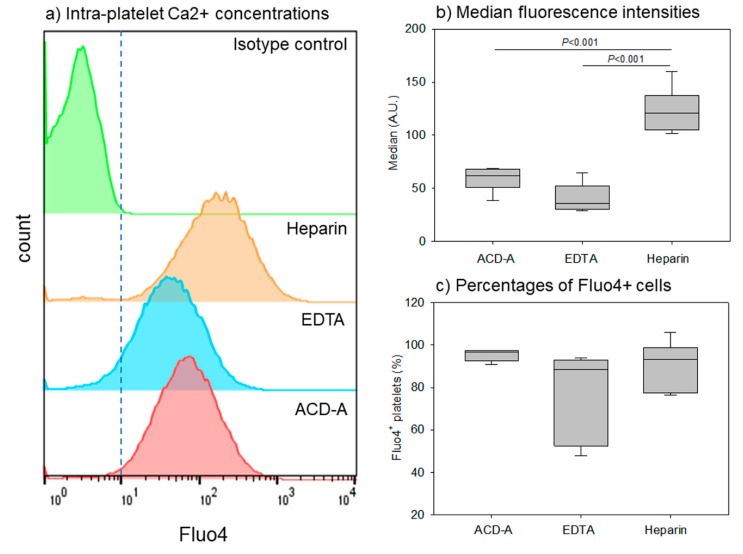
Effects of anticoagulants on intra-platelet free Ca^2+^ concentration. Pure-PRP samples were prepared from whole-blood samples stored for 24 h. (**a**) Representative histograms of Fluo4^+^ platelets. (**b**) Median fluorescence intensities of platelets. (**c**) Percentages of Fluo4^+^ cells. *n* = 6.

**Figure 7 biomedicines-08-00042-f007:**
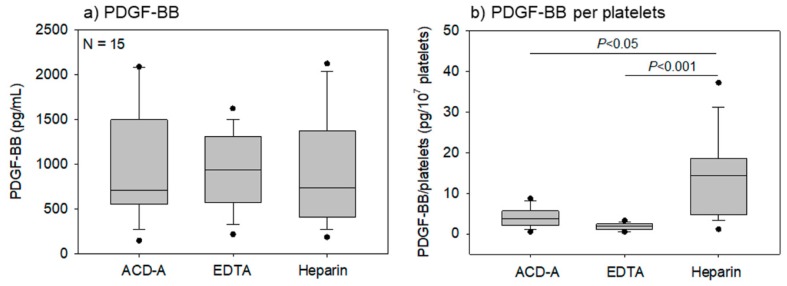
Effects of anticoagulants on Platelet-derived growth factor-BB (PDGF-BB) concentration in pure-PRP samples. Pure-PRP samples were prepared from whole-blood samples stored for 24 h and stored at −80 °C until ELISA. (**a**) PDGF-BB concentration and (**b**) PDGF-BB levels in 10^7^ platelets. *n* = 15.

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
