# Peer review of "A Comparative Study of the Effects of Anticoagulants on Pure Platelet-Rich Plasma Quality and Potency"

_biomedicines, 2020, doi:10.3390/biomedicines8030042_

Round 1

Reviewer 1 Report

Global comments:
This article focuses on the anticoagulant used for preparation of Platelet Rich Plasmas (PRP) for regenerative medicine, and wonders if the EDTA anticoaguulant should be preferable to the conventional one, ACD-A. The authors prepared various PRPs, obtained from 15, non-smoking, healthy volunteers through blood collection using the 3 anticoagulants tested: ACD-A, EDTA and Heparin. Anticoagulant suitability is evaluated through cell counts, platelet and blood cell characteristics, platelet activation during preparation and CaCl2 activatability, and release of growth factors measured as PDGF-BB. Globally the study is well presented, and well-documented with many figures. However, this study is of moderate priority for current practice, as no obvious advantages are evidenced for the replacement of the conventional ACD-A anticoagulant with EDTA, excepted for a better handling of platelets and easier suspension, whilst use of heparin is rejected (as expected). Nevertheless, ACD-A preserves better platelets from activation during the preparation process, and these platelets keep a higher activatability capacity as shown in the study. In addition, the authors do not provide any biological study on the use of the different PRPs obtained with the various anticoagulants tested in regenerative medicine applications.
In addition, it is reminded that not only growth factors can be involved in PRP benefits, but also microparticles released from platelets. Generation and characteristics of extracellular vesicles present with the various anticoagulated PRPs are not discussed in this article. Generation of microparticles (extracellular vesicles) can be impacted by the anticoagulant used.
Authors' comments on suitability of EDTA anticoagulated PRPs opposes to the usual acceptation that ACD-A anticoagulated PRPs suit better the targeted application. A comparative stiudy of the various anticoagulated PRPs in one of tehir applications in regenerative medicine would have been worth.  Probably, this should have provided a stronger evidence on appropriateness of the anticoagulant recommended. I suggest to the authors to include some results on biological activity of PRPs in a regenerative medicine application, if they can have it available.
Concerning methodology, the authors obtained 7 ml of blood (a small volume) using the various anticoagulants, whilst PRP is usually obtained through collection of much larger volumes of blood. This size difference can impact the activation grade of PRPs obtained. Differences reported between anticoagulants could have been different is studies were done using large volumes of blood collection. 
Lastly, integrity of PRPs can be also evaluated by the measurement of Annexin V in plasma supernatant of pelleted platelets. This gives an useful information on the platelet storage lesion, and quality of platelets.

Specific comments:
Some English wording can be improved, especially because some wording is not appropriately used.
On lines 79-80, the authors should indicate why they keep collected blood for  24 h at 22-24 °C. In the results section it becomes obvious, but indicating that they analyzed the variation kinetics of platelets over 24 h is useful at this section.
Lines 80-87, the authors centrifuge the whole blood after 24 h. However, in the results section they show analysis at various blood incubation times. How did they prepare platelets for these studies at different incubation times? This must be indicated in that section.
Line 104, even if referenced, the CaCl2 concentration used must be remindeed here.
Line 115: PBS containing 0.1% BSA should be better.
Lines 116-117: remove the wording repeated twice.
In the results section, the authors analyse platelets after various incubation times. There are 5 analysis points between 0 and 6 h of incubation, and the last analysis is after 24 h, with a gap between 6 h and 24 h. It should be recommended to have testing at 8h, 12 h, 18 h and 24 h, in order to have a more homogeneous analysis of variation kinetics. This is of special relevance for figure 1b, reporting data with the heparin anticoagulant (a sharp variation is noticed between 6 and 24 h, but no other point is available between 6 and 24 h).
Line 134 and beyond: using heparin treatment is not appropriate. It should be better written "heparin anticoagulant".
Legend to figure 2: use "heparin anticoagulant" in place of "heparin treatment".
Line 157 and beyond: using the term of "yield" for platelets, RBCs, and WBCs is not appropriate. The data presented concern platelet count (the objective of the preparation), and the residual contamination with RBCs and WBCs.
Lines 157-158: this is not EDTA tretament but EDTA anticoagulant.
Line 158: WBCs are not concentrated, but they "contaminate" PRP preparation. The same on line 160 for RBCs. 
On legend to figure 3, this is not "yields", but "cell counts".
Line 179 and beyond, the low incidence of CaCl2 on platelet activation is expected, as heparin anticoagulant blocks plasma clotting but not CaCl2 dependent activities, because CaCl2 is not chelated and remains available. These results are then expected.
On figure 7, figures 7b and 7c are not necessary: figure 7b is already presented on figure 3a, and figure 7c gives no information due to the lack of correlation between PDGF-BB and platelet counts, even though there is some "distribution" difference according to the anticoagulant used. Reporting the mean concentrations of PDGF-BB for the 15 PRPs on the 3 different canticoagulants, and SDs, is enough. 
Lines 238-240: the sentence needs to be reformulated.

Author Response

Global comments:

This article focuses on the anticoagulant used for preparation of Platelet Rich Plasmas (PRP) for regenerative medicine, and wonders if the EDTA anticoagulant should be preferable to the conventional one, ACD-A. The authors prepared various PRPs, obtained from 15, non-smoking, healthy volunteers through blood collection using the 3 anticoagulants tested: ACD-A, EDTA and Heparin. Anticoagulant suitability is evaluated through cell counts, platelet and blood cell characteristics, platelet activation during preparation and CaCl2 activatability, and release of growth factors measured as PDGF-BB. Globally the study is well presented, and well-documented with many figures. However, this study is of moderate priority for current practice, as no obvious advantages are evidenced for the replacement of the conventional ACD-A anticoagulant with EDTA, excepted for a better handling of platelets and easier suspension, whilst use of heparin is rejected (as expected). Nevertheless, ACD-A preserves better platelets from activation during the preparation process, and these platelets keep a higher activatability capacity as shown in the study. In addition, the authors do not provide any biological study on the use of the different PRPs obtained with the various anticoagulants tested in regenerative medicine applications.

 Response: As you indicated, we did not provide biological outcomes of EDTA-anticoagulated PRP in an animal study. We admit that this is the weakness. However, to our knowledge, there is no convincing evidence supporting the recommendation that citrate-based anticoagulants are more appropriate than EDTA. We have speculated that this recommendation is based mainly on possible cytotoxic effects of EDTA, in addition to the conventional use of citrate-based anticoagulants in PRP preparation in laboratory testing and surgical procedures. However, we found that neither functional nor morphological damage are produced in platelets by EDTA within 24 h. If platelet membranes are disintegrated, as described elsewhere [e.g., White, Scand J Haemotol, 1968;5:241-54, Yaltirik et al. Platelet-Rich Plasma in Trauma Patients, Trauma in Dentistry, Serdar Gözler, IntechOpen, DOI: 10.5772/intechopen.79966], intra-platelet Ca2+ concentrations could drastically increase and platelet counts could be significantly reduced. However, the data we obtained are quite the opposite to these scenarios. Thus, we suggested that we may re-evaluate EDTA from a “quality” point of view.

In addition, it is reminded that not only growth factors can be involved in PRP benefits, but also microparticles released from platelets. Generation and characteristics of extracellular vesicles present with the various anticoagulated PRPs are not discussed in this article. Generation of microparticles (extracellular vesicles) can be impacted by the anticoagulant used.

 Response: Thank you for your advice. We agree with your idea. In the last year, we have tried to detect platelet-derived microparticles using, mainly using flow-cytometry and fluorescence microscopy. However, because their size is at or below the limits of resolution, we have not yet detected MPs successfully. Thus, unfortunately, we cannot add such data to the revised manuscript.

Authors' comments on suitability of EDTA anticoagulated PRPs opposes to the usual acceptation that ACD-A anticoagulated PRPs suit better the targeted application. A comparative study of the various anticoagulated PRPs in one of their applications in regenerative medicine would have been worth.  Probably, this should have provided a stronger evidence on appropriateness of the anticoagulant recommended. I suggest to the authors to include some results on biological activity of PRPs in a regenerative medicine application, if they can have it available.

 Response: Thank you for your advice. We strongly agree with you. We admit that the lack of animal experiments is the weakness of this study. In-vitro studies are not expected to provide convincing evidence without in-vivo simulated perfusion systems or something similar. However, “ten days” provided for manuscript revision is too short for developing an experimental plan, receiving approval by our university ethics committee, and then performing animal experiments. We hope to use your advice in the next study.

Concerning methodology, the authors obtained 7 ml of blood (a small volume) using the various anticoagulants, whilst PRP is usually obtained through collection of much larger volumes of blood. This size difference can impact the activation grade of PRPs obtained. Differences reported between anticoagulants could have been different is studies were done using large volumes of blood collection. 

 Response: In general, the differences in blood containers, tubes, and bags, are thought to influence PRP quality. Thus, as you indicated, the volume of blood samples may be an essential factor to be considered. However, blood-collection tubes (10 mL or smaller) have been preferably used in dentistry clinical settings. Thus, in this study, we focused on this volume of blood samples for PRP preparation.

Lastly, integrity of PRPs can be also evaluated by the measurement of Annexin V in plasma supernatant of pelleted platelets. This gives a useful information on the platelet storage lesion, and quality of platelets.

 Response: To further investigate the cytotoxic effects of EDTA, as you suggested, the experiment for the detection of apoptosis would be conclusive in evaluating platelet quality. However, assuming that PRP quality can be endorsed by the retention and release of growth factors, it is of questionable value to pursue intact platelets. In addition, because of revision time-limits, we decided not to perform this additional experiment.

Specific comments:

Some English wording can be improved, especially because some wording is not appropriately used.

 Response: The revised manuscript was again edited by professional editing service (Editage). Thus, we hope the revised manuscript meets your standards.

On lines 79-80, the authors should indicate why they keep collected blood for 24 h at 22-24 °C. In the results section it becomes obvious, but indicating that they analyzed the variation kinetics of platelets over 24 h is useful at this section.

 Response: Sample storage depends on the purpose of the study and the study design. In the clinical dentistry setting, PRP is usually prepared on-site from autologous blood samples. Thus, in most cases, the blood samples are immediately or within an hour of collection subjected to PRP preparation. However, for some practical reasons (e.g., the outsourcing of preparation), PRP is also prepared from stored blood samples. Thus, we designed the time-course experiments with incubation for up to 6 h. However, because EDTA did not significantly impair platelets or reduce the amounts of PDGF-BB, we extended the incubation up to 24 h. The amounts of PDGF-BB were slightly decreased after 24 h of incubation, but no substantial damage to platelet functions or morphology was observed. Thus, these data support our original purpose.

 Lines 80-87, the authors centrifuge the whole blood after 24 h. However, in the results section they show analysis at various blood incubation times. How did they prepare platelets for these studies at different incubation times? This must be indicated in that section.

 Response: It is generally thought that EDTA is not appropriate for PRP preparation. Thus, we formed a working hypothesis that EDTA-anticoagulation treatments reduced platelet viability or functions or growth factor contents and investigated these possibilities. Unexpectedly, EDTA did not negatively influence either platelets or growth factors following incubations reflective of clinical settings. Therefore, we extended incubation up to 24 h to further detect any potentially adverse effects of EDTA.

Line 104, even if referenced, the CaCl2 concentration used must be reminded here.

 Response: We added the final concentration (0.1%) there.

Line 115: PBS containing 0.1% BSA should be better.

 Response: For the assays using living platelets, we used 0.1% BSA-containing PBS to minimize platelet aggregation. For flow-cytometric analysis, however, since platelets were fixed, we did not use BSA-containing PBS. In our previous studies [Toyoda et al., Int J Implant Dent 4:23;2018, Kitamura et al., Micron 113:1-9;2018], we examined and confirmed the validity of these experimental conditions.

Lines 116-117: remove the wording repeated twice.

 Response: Thank you for pointing out our careless mistake. We removed this duplication.

In the results section, the authors analyse platelets after various incubation times. There are 5 analysis points between 0 and 6 h of incubation, and the last analysis is after 24 h, with a gap between 6 h and 24 h. It should be recommended to have testing at 8h, 12 h, 18 h and 24 h, in order to have a more homogeneous analysis of variation kinetics. This is of special relevance for figure 1b, reporting data with the heparin anticoagulant (a sharp variation is noticed between 6 and 24 h, but no other point is available between 6 and 24 h).

Response: As described in response to your earlier comment, we simulated the preparation of PRP performed in clinical dentistry setting. In most cases, PRP is prepared from autologous blood samples onsite within an hour. However, sometimes for practical reasons, whole-blood samples are stored for several hours. In the case of PRP preparation outsourcing, whole-blood samples often are stored up to 24 h, but such a scenario is rare.

Initially, we hypothesized that EDTA anticoagulation treatment damages platelets and other blood cells and thereby significantly reduces PRP quality. However, we could not find such adverse effects within 6 h so we extended the incubation to 24 h.

 To our knowledge, heparin has never used for PRP preparation in either PRP studies or PRP therapies because of the mechanism by which it produces anticoagulation. However, heparin is known to preserve growth factors well. Therefore, we adopted heparin as a positive control for growth factor retention studies.

Line 134 and beyond: using heparin treatment is not appropriate. It should be better written "heparin anticoagulant".

 Response: In keeping with your advice, we inserted “anticoagulation” between heparin (or EDTA or ACD-A) and treatments to form “heparin-anticoagulation treatments.”

Legend to figure 2: use "heparin anticoagulant" in place of "heparin treatment".

 Response: As described above, we used “heparin-anticoagulation treatments” there.

Line 157 and beyond: using the term of "yield" for platelets, RBCs, and WBCs is not appropriate. The data presented concern platelet count (the objective of the preparation), and the residual contamination with RBCs and WBCs.

 Response: We rewarded these points.

Lines 157-158: this is not EDTA treatment but EDTA anticoagulant.

 Response: As described above, we used “EDTA-anticoagulation treatments” there.

Line 158: WBCs are not concentrated, but they "contaminate" PRP preparation. The same on line 160 for RBCs.

 Response: We reworded both parts.

On legend to figure 3, this is not "yields", but "cell counts".

 Response: We reworded this point.

Line 179 and beyond, the low incidence of CaCl2 on platelet activation is expected, as heparin anticoagulant blocks plasma clotting but not CaCl2 dependent activities, because CaCl2 is not chelated and remains available. These results are then expected.

 Response: As you mentioned, we think that exogenously added CaCl2 efficiently activates platelets, which were previously treated with Ca2+ chelators, by increasing intra-platelet Ca2+ concentrations. Therefore, because heparin has no direct effects on intra-platelet Ca2+ concentrations, we did not expect such a CaCl2-induced platelet activation in heparin-anticoagulated samples. However, heparin somehow increased basal Ca2+ levels. Thus, we also examined the effects of CaCl2 on heparin-anticoagulated platelets.

Our primary focus is “ACD-A vs. EDTA,” but these heparin data (e.g., the increases in the basal levels of CD62P expression) may provide additional evidence that heparin is inappropriate for PRP preparation.

On figure 7, figures 7b and 7c are not necessary: figure 7b is already presented on figure 3a, and figure 7c gives no information due to the lack of correlation between PDGF-BB and platelet counts, even though there is some "distribution" difference according to the anticoagulant used. Reporting the mean concentrations of PDGF-BB for the 15 PRPs on the 3 different anticoagulants, and SDs, is enough.

 Response: According to your advice, we have deleted panels b and c in Figure 7. However, we thought that panel a facilitates the understanding that the total amounts of PDGF-BB were very similar among PRP prepared using different anticoagulants. Thus, we modified Figure 7 by deleting panels b and c.

Lines 238-240: the sentence needs to be reformulated.

 Response: This sentence has been restructured in the revised manuscript.

Reviewer 2 Report

Comments to the Authors

General comments:

The use of PRP is expending as a therapy in regenerative medicine and while it is not FDA-approved in the US, it can be legally used ‘off-label’. However, the lack of standardization of the techniques is a well-recognized problems that invariably leads to inconsistent quality of PRP among practices worldwide. The multifactorial parts of the methodologies could be a cause of high variability between large and small practices. One of them being the effect of anticoagulants regarding the preparation of PRP and is the focus of the authors in the present study. It is well established that EDTA is best for preserving cell morphology, citrate for platelet function and ACD-A for transfusion-based strategies, and that heparin prevents clotting but could interferes with platelet aggregation thus reducing platelet number. The effect of anticoagulant for blood collection in regenerative medicine has been deemed minor and has already been addressed by many authors [e. g..Ref 1 and Ref 2.]. Moreover, the effect of centrifugation, tubes, platelet resuspension and other individual differences during this process or the use of kits in the preparation of PRP certainly play a role with either of the anticoagulants used to collect the blood. That cannot be assessed simply but may be an aspect that could be discussed further.

The study as presented deals with in-vitro characteristics of the PRP but do not discuss its clinical implication and relevance. The efficacy of PRP used in regenerative medicine is mostly dependent on the release and quality of growth factors and has been reported by many authors [e.g.Ref 3 and Ref 4]. This should be more extensively discussed.

In addition to this, all reported studies used healthy donors but people ongoing regenerative medicine may be under medications that could interfere with PRP preparation.  Another aspect to discuss to broaden the scope of the manuscript.

Specific to the study

The anticoagulant at collection is only on aspect of the study, platelet count and hematocrit that are different for each donor could change the platelet yield. Due to this variation among donors, all the results in this study and their concomitant analysis will benefit from the normalization of the data at baseline.

The 24 h storage time for the evaluation in Sections 2.3 to 2.5 should be indicated in methods and this aspect of storage should be discussed.

PDGF reported to platelet count (Fig7) was examined in this manuscript; however, an analysis of CD62p expression and PDGF following stimulation by CaCl2++ would have been interesting.

In discussion

L 228 References 17,27,28 do not seem to refer to previous studies from these authors.

Citrate as well as EDTA are calcium chelator. The remark L 234-240 is not justified.

Also, note that EDTA is the preferred anticoagulant for assessing CBC. The final aspect of the PRP is not only dependent on the anticoagulant but also from the resuspension method.

Last paragraph L252, viability of platelets does not seem to be an issue for this article, please specify in the introduction. Then, the true role of the anticoagulants for this preparation need to be clarify. Different platelet bio-products and their application have been defined [Ref 5], so may be the preparation should be tailor-made for the final product. Again, this could be a point of discussion.

There is no discussion about the clinical relevance of these in vitro findings.

Ref 1: Graiet H, et al. Use of platelet-rich plasma in regenerative medicine: technical tools for correct quality control. BMJ Open Sport Exerc Med. 2018 Nov 13;4(1)

Ref 2: Lei H1, Gui L, Xiao R. The effect of anticoagulants of the quality and biological efficacy of platelet-rich plasma. Clin Biochem. 2009 Sep;42(13-14):1452-60

Ref 3: Magalon J, et al.  Characterization and comparison of 5 platelet-rich plasma preparations in a single-donor model.  Arthroscopy. 2014 May;30(5):629-38;

Ref 4: do Amaral RJ, et al. Platelet-Rich Plasma Obtained with Different Anticoagulants and Their Effect on Platelet Numbers and Mesenchymal Stromal Cells Behavior In Vitro. Stem Cells Int. 2016;2016:7414036

Ref 5: Acebes-Huertaa,A. Platelet-derived bio-products: Classification update, applications, concerns and new perspectives. Transfus Apher Sci. 2019 Dec 31:102716

Author Response

General comments:

The use of PRP is expending as a therapy in regenerative medicine and while it is not FDA-approved in the US, it can be legally used ‘off-label’. However, the lack of standardization of the techniques is a well-recognized problems that invariably leads to inconsistent quality of PRP among practices worldwide. The multifactorial parts of the methodologies could be a cause of high variability between large and small practices. One of them being the effect of anticoagulants regarding the preparation of PRP and is the focus of the authors in the present study. It is well established that EDTA is best for preserving cell morphology, citrate for platelet function and ACD-A for transfusion-based strategies, and that heparin prevents clotting but could interferes with platelet aggregation thus reducing platelet number. The effect of anticoagulant for blood collection in regenerative medicine has been deemed minor and has already been addressed by many authors [e. g. Ref 1 and Ref 2.]. Moreover, the effect of centrifugation, tubes, platelet resuspension and other individual differences during this process or the use of kits in the preparation of PRP certainly play a role with either of the anticoagulants used to collect the blood. That cannot be assessed simply but may be an aspect that could be discussed further.

 Response: As you indicated, we also posit that PRP quality is positively or negatively influenced by many factors, such as centrifugal conditions, types of blood-collection tubes, skills of pipetting, among other factors. To standardize future preparation protocols, these factors should be investigated step-by-step and reevaluated to advance our understanding of PRP.

As far as we searched, in addition to those publications, Araki et al. [Tissue Eng 2012, 18:176] extensively investigated EDTA as an anticoagulant for PRP preparation. However, these studies did not assess biological outcomes in animal experiments. do Amaral et al. [Stem Cell Int 2016, ID7414036] demonstrated the effects of EDTA-anticoagulated PRP preparations in the in-vitro experimental system. However, without physiological diffusion and exclusion systems, it can be predicted that prolonged exposure of fibroblastic or similar cells to EDTA results in impaired cell proliferation. Therefore, we propose that we should further investigate the clinical applicability of EDTA-anticoagulated PRP preparations in animal studies. This limitation is added with our opinion at the end of the Discussion.

The study as presented deals with in-vitro characteristics of the PRP but do not discuss its clinical implication and relevance. The efficacy of PRP used in regenerative medicine is mostly dependent on the release and quality of growth factors and has been reported by many authors [e.g. Ref 3 and Ref 4]. This should be more extensively discussed.

 Response: In the third paragraph of the Discussion, we added extensive discussion of this matter.

In addition to this, all reported studies used healthy donors but people ongoing regenerative medicine may be under medications that could interfere with PRP preparation. Another aspect to discuss to broaden the scope of the manuscript.

 Response: Thank you for this comment. We agree with you and we think it is much more important to investigate the effects of medication on the following PRP therapy. However, without basic knowledge under physiological conditions, it is challenging to design a study and determine experimental conditions that simulate medicated patients. We posit that such a trial may not be able to obtain clear evidence or reach convincing conclusions easily. In addition, expansion of the discussion to include such considerations may render the narrative less focused. Thus, we chose not to broaden the scope of the manuscript but will consider these issues in future investigations.

Specific to the study

The anticoagulant at collection is only on aspect of the study, platelet count and hematocrit that are different for each donor could change the platelet yield. Due to this variation among donors, all the results in this study and their concomitant analysis will benefit from the normalization of the data at baseline.

 Response: Individual patient variations are a potentially confounding factor in PRP and many other areas of research. While without further study we cannot speculate whether our proposed solution is sufficient to normalize preparations to encompass all potential individual variations, we propose that such normalization is a necessary step to assure quality in PRP therapy.

The 24 h storage time for the evaluation in Sections 2.3 to 2.5 should be indicated in methods and this aspect of storage should be discussed.

 Response: We added the storage time in each section.

PDGF reported to platelet count (Fig7) was examined in this manuscript; however, an analysis of CD62p expression and PDGF following stimulation by CaCl2++ would have been interesting.

 Response: According to the comment of Reviewer 1, we have modified Figure 7 by deleting panels b and c.

As described above, we posit that not only the quantity but also the quality of platelets are key criteria in assuring PRP quality. Thus, we evaluated the CD62P expression in this study. In our previous study [Tsujino et al., Dent J, 2019, 7:E61], we developed and reported a simple spectrophotometric method for evaluation of platelet aggregation activity.

In discussion

L 228 References 17,27,28 do not seem to refer to previous studies from these authors.

 Response: Thank you for indicating our careless mistake; we reworded this sentence accordingly.

Citrate as well as EDTA are calcium chelator. The remark L 234-240 is not justified.

 Response: As indicated, both EDTA and citrate function as an anticoagulant by chelating calcium. However, these agents can be distinguished from each other by their binding ability. According to Keowmaneechai and McClements [J Agric Food Chem. 2002, 20;50:7145-53], EDTA and citrate both bound calcium ions in a 1:1 ratio, but EDTA had a much higher binding constant. Thus, we posit that EDTA has previously been considered inappropriate for PRP preparation in terms of the preservation of intact platelets.

Also, note that EDTA is the preferred anticoagulant for assessing CBC. The final aspect of the PRP is not only dependent on the anticoagulant but also from the resuspension method.

 Response: As indicated, because of its ability to minimize platelet aggregation and platelet-leukocyte binding, EDTA has been recommended for assessing CBCs. Also, in PRP preparation, EDTA-anticoagulated blood samples do not require highly developed pipetting skills. Because this is undoubtedly a significant advantage for PRP users, we proposed a reconsideration of EDTA as an anticoagulant choice.

However, at the same time, it is essential to precisely evaluate the potential adverse effects of any remaining anticoagulants in individual PRP preparations on wound healing and tissue regeneration. For example, from a pharmacokinetic point of view, we should know how local concentrations of EDTA contained in PRP preparations change with time, and how EDTA influences tissues at implantation sites prior to any clinical application of EDTA.

Last paragraph L252, viability of platelets does not seem to be an issue for this article, please specify in the introduction. Then, the true role of the anticoagulants for this preparation need to be clarify. Different platelet bio-products and their application have been defined [Ref 5], so may be the preparation should be tailor-made for the final product. Again, this could be a point of discussion.

 Response: To our knowledge, there is no convincing explanation as to why EDTA is not preferable for PRP preparation. However, because of its higher chelating power, we speculate that concerns that EDTA may induce unexpected adverse effects, such as disruption of platelets and leukocytes including those included in PRP preparations, or potential interference in tissue regeneration may account for this. Thus, at the beginning of this study, we aimed to obtain direct evidence that EDTA anticoagulation impairs the quality and quantity of platelets by strongly chelating calcium ions. However, EDTA activated platelets and slightly reduced intra-platelet Ca2+ concentrations but did not significantly reduce platelet counts or growth factor content.

Therefore, we had to modify our working hypothesis as below. EDTA may be used for Therefore, we modified our working hypothesis, as stated below. EDTA may be used for PRP preparation without any significant adverse effects on tissue regeneration. As described in many publications, PRP is used as a carrier in growth factor cocktails for regenerative therapy. In this context, the amount of growth factors is the most crucial point for evaluation. Our data regarding PDGF-BB content suggests that PRP preparations prepared from EDTA-anticoagulated blood samples can be stored for up to 24 h. Therefore, we concluded that although further biological assay of EDTA-anticoagulated PRP preparations is required, a broader re-assessment of EDTA as an anticoagulant for PRP preparation should be considered.

Conversely, PRP preparations prepared without anticoagulants, which are often designated “injectable-PRF (Choukroun)” or “liquid-PRF (Miron),” have been investigated recently. These investigators proposed that PRP preparations containing no anticoagulants are much safer and suggest this type of PRP preparations for regenerative therapy as a promising alternative to conventional types of anticoagulated PRP preparations. This preparation protocol requires operators and clinicians to prepare PRP promptly, usually within a timeframe as short as 30 min from blood collection, but we predict that this type of PRP may become more popular in the next five years.  

We suggest it is difficult to define the “true” roles of anticoagulants in PRP preparations. There is no doubt that operators and clinicians are given flexibility in the timing of PRP preparation by anticoagulants. However, except for this advantage, we cannot currently speculate regarding their “true” advantage (role) over any potential biological disadvantages without further study.

There is no discussion about the clinical relevance of these in vitro findings.

 Response: As briefly described in the abstract, EDTA treatment seems to simplify the process of PRP preparation. In citrated or ACD-anticoagulated blood samples, the pipetting skills of the preparer influence the quality of PRP preparations, including platelet counts. Thus, we propose that the choice of EDTA may contribute to improved quality control by reducing low-quality PRP preparations. However, we should examine and clarify the possible adverse effects of EDTA on tissue regeneration when included in PRP preparations prior to its clinical use. Thus, given these unknowns, we hesitate to expand the discussion regarding the clinical relevance further.

Ref 1: Graiet H, et al. Use of platelet-rich plasma in regenerative medicine: technical tools for correct quality control. BMJ Open Sport Exerc Med. 2018 Nov 13;4(1)

Ref 2: Lei H1, Gui L, Xiao R. The effect of anticoagulants of the quality and biological efficacy of platelet-rich plasma. Clin Biochem. 2009 Sep;42(13-14):1452-60

Ref 3: Magalon J, et al.  Characterization and comparison of 5 platelet-rich plasma preparations in a single-donor model.  Arthroscopy. 2014 May;30(5):629-38;

Ref 4: do Amaral RJ, et al. Platelet-Rich Plasma Obtained with Different Anticoagulants and Their Effect on Platelet Numbers and Mesenchymal Stromal Cells Behavior In Vitro. Stem Cells Int. 2016;2016:7414036

Ref 5: Acebes-Huertaa,A. Platelet-derived bio-products: Classification update, applications, concerns and new perspectives. Transfus Apher Sci. 2019 Dec 31:102716

Reviewer 3 Report

The article is of scientific interest. The research was performed qualitatively. The article is well written. There is no prospect of practical use.

Author Response

The article is of scientific interest. The research was performed qualitatively. The article is well written. There is no prospect of practical use.

Response: Thank you for your comments. As advised by other reviewers, we proposed further testing of different PRP preparations using various anticoagulants for their practical use in biological studies.

Reviewer 4 Report

The paper from Aizawa et al. analyses the effect of different anti-coagulants in the PRP preparation, on several parameters of platelets. Indeed, the choice of anti-coagulant surely could be one of the parameters to be considered in the preparation of platelet derivatives in order to maintain the greatest biological activity to stimulate wound healing (along with platelet concentration, presence/absence of leukocytes, devices…). In this point of view, the paper deals with a very trending topic in the regenerative medicine.

The paper, on the whole, is quite clear and the results very interesting but, in this reviewer’s opinion, some aspects could be improved.

The introduction is well written, concise and clear. The authors suggest some hypothesis to explain “different or unexpected clinical data” not mentioning that has been widely demonstrated that the concentration of platelets in PRP (at least in vitro but it is not to be excluded in vivo too) could deeply affect the biological outcome in terms of regeneration and healing (see for example “Giusti I et al. The effects of platelet gel-released supernatant on human fibroblasts. Wound Repair Regen. 2013;21(2):300-8” or “Giusti I et al. In vitro evidence supporting applications of platelet derivatives in regenerative medicine. Blood Transfus. 2019:1-12”). It could be appropriate for the authors to briefly develop this topic too.

 The experimental section can be improved in some parts.

Paragraph 2.1 It is not clear (line 80) if platelets have been stored at room temperature under agitation or not. Agitation is a fundamental parameter influencing platelet aggregation. Please clarify. Paragraph 2.5. It is not clear if platelet have been activated prior to perform the ELISA in order to allow the release of PDGF-BB: if yes, please specify; if not, please explain this choice. Moreover, it could be advisable to add some details on the procedure used, for example if samples needed to be diluted.

Regarding the Results, this review has some concerns that need to be clarified.

Figure 1: authors affirm that heparin treatment significantly reduced platelet counts and white blood cell counts. Do they mean statistically significant? If positive, they should show that graphically on the figures and add in the legend the P resulting from statistical analysis. If not, how one can suppose by the big error bars, the term “significantly” is completely misleading and should not be used.

The discussion needs to be “discussed” more widely.

For example, in Figure 2 and figure 4, how do the authors explain the change in platelet size induced by heparin and EDTA? What should be the mechanism by which an anticoagulant affects the platelets’ volume?

Moreover, since the release of growth factors from platelets relies on their activation, did the authors verify if the different anti-coagulants could differently impact on/impair the following activation step?

Some typos:

Line 81: correct 1st soft spin in 1st soft spin

Line 83: correct 2nd hard spin in 2nd hard spin

Line 104. Correct CaCl2 into CaCl2

Line 116: the sentence “treated with 5 μg/mL Fluo4-AM (Dojin, Kumamoto, Japan)” is repeated twice

Line 109, line 141, line 153: “ambient temperature” could be replaced by a more usual “room temperature”

Line 212: correct 107 into 107

Line 240: “EDTA may be used PRP preparation” should be replaced with “EDTA may be used in PRP preparation”.

Author Response

The paper from Aizawa et al. analyses the effect of different anti-coagulants in the PRP preparation, on several parameters of platelets. Indeed, the choice of anti-coagulant surely could be one of the parameters to be considered in the preparation of platelet derivatives in order to maintain the greatest biological activity to stimulate wound healing (along with platelet concentration, presence/absence of leukocytes, devices…). In this point of view, the paper deals with a very trending topic in the regenerative medicine.

The paper, on the whole, is quite clear and the results very interesting but, in this reviewer’s opinion, some aspects could be improved.

The introduction is well written, concise and clear. The authors suggest some hypothesis to explain “different or unexpected clinical data” not mentioning that has been widely demonstrated that the concentration of platelets in PRP (at least in vitro but it is not to be excluded in vivo too) could deeply affect the biological outcome in terms of regeneration and healing (see for example “Giusti I et al. The effects of platelet gel-released supernatant on human fibroblasts. Wound Repair Regen. 2013;21(2):300-8” or “Giusti I et al. In vitro evidence supporting applications of platelet derivatives in regenerative medicine. Blood Transfus. 2019:1-12”). It could be appropriate for the authors to briefly develop this topic too.

 Response: In the Introduction, we added a brief discussion of the clinical use of PRP supported by the data obtained from in vitro studies by citing these articles as well as ours.

However, regarding the correlation between platelet counts and biological outcomes, some influential researchers disagree and dispute whether such correlations are valid. Thus, although we can claim the importance of accurate platelet counting (as an advantage of EDTA anticoagulation) for more predictable PRP therapy on such correlation in constructing the narrative of this article based, we decided not to mention it to avoid their strong rebuttal.

The experimental section can be improved in some parts.

Paragraph 2.1 It is not clear (line 80) if platelets have been stored at room temperature under agitation or not. Agitation is a fundamental parameter influencing platelet aggregation. Please clarify. Paragraph 2.5. It is not clear if platelet have been activated prior to perform the ELISA in order to allow the release of PDGF-BB: if yes, please specify; if not, please explain this choice. Moreover, it could be advisable to add some details on the procedure used, for example if samples needed to be diluted.

 Response: We used the term “inverting” in the sense of “agitation.” However, in the revised manuscript, we replaced this term with “agitation.”

We did not activate platelets before freezing them in this study; however, we preliminarily confirmed that the freeze-thawing process could release growth factors stored in platelets. In addition, we added that we diluted the samples with PBS to fit the PDGF-BB concentrations into the range of the ELISA calibration curve.

Regarding the Results, this review has some concerns that need to be clarified.

Figure 1: authors affirm that heparin treatment significantly reduced platelet counts and white blood cell counts. Do they mean statistically significant? If positive, they should show that graphically on the figures and add in the legend the P resulting from statistical analysis. If not, how one can suppose by the big error bars, the term “significantly” is completely misleading and should not be used.

 Response: We performed statistical analyses as described in the Materials and Methods (subsection 2.6). In the revised manuscript, according to your comment, we modified the figure and the legend.

The discussion needs to be “discussed” more widely.

For example, in Figure 2 and figure 4, how do the authors explain the change in platelet size induced by heparin and EDTA? What should be the mechanism by which an anticoagulant affects the platelets’ volume?

 Response: We propose that heparin-induced enlargement is probably due to the aggregation of several platelets (~10 platelets). In support of this hypothesis, platelet counts are reduced in heparin-anticoagulated blood samples. Thus, we posit that these apparent changes may be mainly due to an artifact of the measurement method based on the Coulter principle.

Conversely, an earlier study (White, Scandinavian Journal of Haematology, 1968;5:241-54), suggested that the chelation of membrane calcium by EDTA appears to produce marked irregularities in platelet walls and massive swelling of the canalicular system. This author also mentioned that swelling of internal channels is the dominant alteration in EDTA-treated platelets. Thus, we posit that hypotonic conditions produced on the platelet membrane surface trigger the swelling of individual platelets. Since to our knowledge, there is no recent publication that excludes or refutes this possibility, we did not expand this discussion in the revised manuscript.

Moreover, since the release of growth factors from platelets relies on their activation, did the authors verify if the different anti-coagulants could differently impact on/impair the following activation step?

 Response: To address this, we considered two points, namely, the basal levels and the increases in activation markers. As described in this study, compared with ACD-A, EDTA, and heparin upregulated CD62P expression in resting control platelets. To our knowledge, the mechanisms behind these phenomena have not been thoroughly investigated to date. We speculate that changes in platelet morphology may result in platelet activation.

Conversely, EDTA and heparin reduced the CaCl2-induced increases in CD62P expression. Thus, the increases in CD62P expression (-fold increase) were the highest following ACD-A treatment and the lowest following heparin treatment. Because heparin is not a Ca2+ chelator, heparin-anticoagulated platelets may be less sensitive to exogenously added CaCl2.

In clinical settings, thrombin is often used for activation in addition to CaCl2. In heparin-anticoagulated PRP, it is not expected that thrombin induces clot formation. In contrast, judging from their mechanisms, it can be concluded that neither ACD-A and EDTA impact clot formation.

Some typos:

Line 81: correct 1st soft spin in 1st soft spin

 Response: We corrected this.

  Line 83: correct 2nd hard spin in 2nd hard spin

 Response: We corrected this.

Line 104. Correct CaCl2 into CaCl2

 Response: We corrected this.

Line 116: the sentence “treated with 5 μg/mL Fluo4-AM (Dojin, Kumamoto, Japan)” is repeated twice

 Response: We removed this duplication.

Line 109, line 141, line 153: “ambient temperature” could be replaced by a more usual “room temperature”

 Response: We replaced with “ambient temperature” with “room temperature.”

Line 212: correct 107 into 107

 Response: We corrected this.

Line 240: “EDTA may be used PRP preparation” should be replaced with “EDTA may be used in PRP preparation”.

 Response: According to the other reviewer’s suggestion (reviewer 1), we slightly modified this sentence.

Round 2

Reviewer 2 Report

Comments to the Authors

Thank you for clarifying the text of this manuscript. Despite justification on specific points some aspects have not been approached.

In the introduction, the ultimate quality of PRP for clinical application should be clearly defined. What in-vitro characterization is relevant for the effectiveness of PRP?  Is it the yield of extraction of growth factor, the platelet function, should platelet be intact or others markers that are important factors for the clinical application?

In response to: “The study as presented deals with in-vitro characteristics of the PRP but do not discuss its clinical implication and relevance. The efficacy of PRP used in regenerative medicine is mostly dependent on the release and quality of growth factors This should be more extensively discussed” you mentioned: In the third paragraph of the Discussion, we added extensive discussion of this matter.

However, if it is the extraction yield of growth factor then this manuscript does not provide strong evidence that EDTA supports this.

In your response to  “Due to this variation among donors, all the results in this study and their concomitant analysis will benefit from the normalization of the data at baseline”  you recognized that:  Individual patient variations are a potentially confounding factor in PRP and many other areas of research. While without further study we cannot speculate whether our proposed solution is sufficient to normalize preparations to encompass all potential individual variations, we propose that such normalization is a necessary step to assure quality in PRP therapy.

 Could you make an attempt to normalize results based on the initial data of the donation? No further studies are necessary to answer this question.

“An analysis of CD62p expression and PDGF following stimulation by CaCl2++ would have been interesting”

You responded: Thus, we evaluated the CD62P expression in this study.

The relationship between CD62p and PDGF was anticipated as an answer (or a graph).

To your last comment regarding the preparation of PRP, I would add that a resting period is necessary before resuspension of platelets. This is not mentioned in the material and method and could be a major factor for platelet function. Pipetting as you mentioned could affect the end results. May be a comment regarding the volume to be resuspended could be added.

Author Response

Thank you for clarifying the text of this manuscript. Despite justification on specific points some aspects have not been approached.

1) In the introduction, the ultimate quality of PRP for clinical application should be clearly defined. What in-vitro characterization is relevant for the effectiveness of PRP?  Is it the yield of extraction of growth factor, the platelet function, should platelet be intact or others markers that are important factors for the clinical application?

Response: Even though this question is pertinent to our study, it remains to be clearly addressed, especially in terms of factors influencing the PRP quality. However, to respond to your comment, we added our definition of PRP quality to the Introduction section.

To the best of our knowledge, there have been no convincing explanations for PRP quality; however, without such a sense of manufacturing, we will probably never improve the current PRP therapy in terms of predictability. In fact, since we proposed the necessity for standardization of preparation protocols [Kobayashi et al., Biologicals 40:323-329; 2012], several researchers have increasingly proposed various protocols for standardization, seemingly in response to our initiative.

However, the protocol standardization does not completely assure the quality of individual PRP samples. To ensure PRP quality, we should inspect individual products, that is, PRP samples, before use. If not, we should at least conduct all product inspections immediately after application to “normalize” the clinical outcomes to make the randomized clinical trial results more convincing. As reported in our previous studies [Kawase and Okuda K. BioMed Res Int, Volume 2018, Article ID 6389157, Kawase et al., Int J Growth Factors Stem Cells Dent 2:13-17; 2019, Kitamura et al., Int J Implant Dent 4:29; 2018, Tsujino et al., Dent J 7(2):61; 2019], we have persistently stressed on the necessity for quality inspection. Concurrently, Milants et al. proposed novel classifications of PRP on the basis of blood cell content and activation status [BioMed Res Int, Volume 2017, Article ID 7538604]. Such classifications are also expected to be useful for quality evaluation. However, the current status is not yet sufficient for realizing a standard PRP therapy with predictable results. Achievement of these goals warrants more time.

At present, we have focused on the quantity of growth factors, platelet function, and intactness as key factors influencing the clinical outcomes of PRP therapy. However, once we achieve this goal in the near future, the yield of growth factor extraction, platelet function, intactness of platelet, or other marker candidates may no longer be considered important factors for PRP quality. However, at present, there is no evidence that would lead to the exclusion of our proposal for the possible factors influencing PRP quality.

2) In response to: “The study as presented deals with in-vitro characteristics of the PRP but do not discuss its clinical implication and relevance. The efficacy of PRP used in regenerative medicine is mostly dependent on the release and quality of growth factors This should be more extensively discussed” you mentioned: In the third paragraph of the Discussion, we added extensive discussion of this matter.

However, if it is the extraction yield of growth factor then this manuscript does not provide strong evidence that EDTA supports this.

Response: As mentioned previously, we do not strongly recommend the use of EDTA for PRP preparation in this study. Instead, we wanted to reconsider the suitability of EDTA for PRP preparation. To date, it has generally been considered that EDTA disrupts the plasma membrane of platelets as well as that of other cell types. In addition, remnants of EDTA in PRP preparations may negatively affect cells and tissues surrounding the implantation site. Therefore, EDTA is not recommended for PRP preparation [Harrison, J Thromb Haemost. 2018]. However, as we revealed in this study, platelets were not significantly disrupted, at least not within 24 h, when the factory-made powder EDTA-containing blood collection tubes were used. In addition, PDGF-BB content did not decrease significantly in the resulting PRP preparations, although platelets were activated in the EDTA-anticoagulated blood samples. Therefore, considering its advantage in platelet suspension preparation, we concluded that EDTA may not be unsuitable for this purpose and should be reevaluated.

3) In your response to “Due to this variation among donors, all the results in this study and their concomitant analysis will benefit from the normalization of the data at baseline” you recognized that: Individual patient variations are a potentially confounding factor in PRP and many other areas of research. While without further study we cannot speculate whether our proposed solution is sufficient to normalize preparations to encompass all potential individual variations, we propose that such normalization is a necessary step to assure quality in PRP therapy.

Could you make an attempt to normalize results based on the initial data of the donation? No further studies are necessary to answer this question.

Response: We agree with your suggestion. We have always considered such a possibility. However, we are yet to devise a clear idea for normalization.

4) “An analysis of CD62p expression and PDGF following stimulation by CaCl2++ would have been interesting”

You responded: Thus, we evaluated the CD62P expression in this study. The relationship between CD62p and PDGF was anticipated as an answer (or a graph).

Response: Thank you for this comment. It is generally considered that CD62P expression is related to exocytosis of alpha-granules and other types of granules stored in platelets, that is, inversion of the plasma membrane, in the process of activation in response to external stimuli. Therefore, we inferred that upregulated CD62P expression and PDGF-BB release occur simultaneously, as demonstrated in our previous study [Tsujino et al., Dent J 7(4):109; 2019]. However, under the experimental conditions in this study, since PDGF-BB is detected in freeze-thawed samples regardless of its state, such as “released from” or “stored in” platelets, we did not expect the possible positive correlation between CD62P expression and total PDGF-BB content in PRP samples. It is attributed to technical limitations.

5) To your last comment regarding the preparation of PRP, I would add that a resting period is necessary before resuspension of platelets. This is not mentioned in the material and method and could be a major factor for platelet function. Pipetting as you mentioned could affect the end results. May be a comment regarding the volume to be resuspended could be added.

Response: We did not introduce any intervals, such as resting periods, between centrifugation and the final resuspension, or between the 1st and 2nd centrifugations. Unfortunately, we do not have evidence to validate our procedure or this comment; however, we were apprehensive of tight platelet aggregation in case of ACD-A when platelets are incubated in the form of pellets for longer time periods.

As for the volume, as described in Line 94, platelet pellets were resuspended in 250 μL of PPP (platelet-poor plasma) for all the samples.

Reviewer 4 Report

All suggestions have been widely addressed, clarifying many doubts.

In paragraph 2.5 it has been added that samples were diluted in PBS to perform ELISA, which is totally trustworthy. Please, add the dilution factor too. 

In lines 212-213 it is affirmed "Effects of anticoagulants on PDGF-BB concentration in pure-PRP samples are shown in Figure 7" that seems to be in contradiction with what affirmed above. Please rephrase the sentence. Do the authors  mean that the reported value have been corrected accordingly to the dilution factor?

Author Response

All suggestions have been widely addressed, clarifying many doubts.

1) In paragraph 2.5 it has been added that samples were diluted in PBS to perform ELISA, which is totally trustworthy. Please, add the dilution factor too.

Response: Thank you for your advice. Only in the ACD-A-anticoagulated samples, PDGF-BB levels were diluted by 1.2-foldin whole-blood samples. Furthermore, the resulting PRP preparations were diluted by a factor of 50 with PBS.

2) In lines 212-213 it is affirmed "Effects of anticoagulants on PDGF-BB concentration in pure-PRP samples are shown in Figure 7" that seems to be in contradiction with what affirmed above. Please rephrase the sentence. Do the authors mean that the reported value have been corrected accordingly to the dilution factor?

Response: We rephrased this sentence by adding the phrase “corrected by dilution.”

Round 3

Reviewer 2 Report

Comments to the Authors

Thank you for your comments and clarification.

I would like to see some of your response below to be integrated in either the introduction and/or discussion

To the best of our knowledge, there have been no convincing explanations for PRP quality; however, without such a sense of manufacturing, we will probably never improve the current PRP therapy in terms of predictability. In fact, since we proposed the necessity for standardization of preparation protocols [Kobayashi et al., Biologicals 40:323-329; 2012], several researchers have increasingly proposed various protocols for standardization, seemingly in response to our initiative.

However, the protocol standardization does not completely assure the quality of individual PRP samples. To ensure PRP quality, we should inspect individual products, that is, PRP samples, before use. If not, we should at least conduct all product inspections immediately after application to “normalize” the clinical outcomes to make the randomized clinical trial results more convincing. As reported in our previous studies [Kawase and Okuda K. BioMed Res Int, Volume 2018, Article ID 6389157, Kawase et al., Int J Growth Factors Stem Cells Dent 2:13-17; 2019, Kitamura et al., Int J Implant Dent 4:29; 2018, Tsujino et al., Dent J 7(2):61; 2019], we have persistently stressed on the necessity for quality inspection. Concurrently, Milants et al. proposed novel classifications of PRP on the basis of blood cell content and activation status [BioMed Res Int, Volume 2017, Article ID 7538604]. Such classifications are also expected to be useful for quality evaluation. However, the current status is not yet sufficient for realizing a standard PRP therapy with predictable results. Achievement of these goals warrants more time.

At present, we have focused on the quantity of growth factors, platelet function, and intactness as key factors influencing the clinical outcomes of PRP therapy. However, once we achieve this goal in the near future, the yield of growth factor extraction, platelet function, intactness of platelet, or other marker candidates may no longer be considered important factors for PRP quality. However, at present, there is no evidence that would lead to the exclusion of our proposal for the possible factors influencing PRP quality.

Author Response

Reviewer 2

Thank you for your comments and clarification.

I would like to see some of your response below to be integrated in either the introduction and/or discussion.

To the best of our knowledge, there have been no convincing explanations for PRP quality; however, without such a sense of manufacturing, we will probably never improve the current PRP therapy in terms of predictability. In fact, since we proposed the necessity for standardization of preparation protocols [Kobayashi et al., Biologicals 40:323-329; 2012], several researchers have increasingly proposed various protocols for standardization, seemingly in response to our initiative.

However, the protocol standardization does not completely assure the quality of individual PRP samples. To ensure PRP quality, we should inspect individual products, that is, PRP samples, before use. If not, we should at least conduct all product inspections immediately after application to “normalize” the clinical outcomes to make the randomized clinical trial results more convincing. As reported in our previous studies [Kawase and Okuda K. BioMed Res Int, Volume 2018, Article ID 6389157, Kawase et al., Int J Growth Factors Stem Cells Dent 2:13-17; 2019, Kitamura et al., Int J Implant Dent 4:29; 2018, Tsujino et al., Dent J 7(2):61; 2019], we have persistently stressed on the necessity for quality inspection. Concurrently, Milants et al. proposed novel classifications of PRP on the basis of blood cell content and activation status [BioMed Res Int, Volume 2017, Article ID 7538604]. Such classifications are also expected to be useful for quality evaluation. However, the current status is not yet sufficient for realizing a standard PRP therapy with predictable results. Achievement of these goals warrants more time.

At present, we have focused on the quantity of growth factors, platelet function, and intactness as key factors influencing the clinical outcomes of PRP therapy. However, once we achieve this goal in the near future, the yield of growth factor extraction, platelet function, intactness of platelet, or other marker candidates may no longer be considered important factors for PRP quality. However, at present, there is no evidence that would lead to the exclusion of our proposal for the possible factors influencing PRP quality.

Response: Thank you for this advice. We have modified the 2nd paragraph of the Introduction section and inserted an additional paragraph at the end of the Discussion section to complementarily express our thoughts regarding PRP quality.